# Semi-Volatile Organic Compounds in Car Dust: A Pilot Study in Jeddah, Saudi Arabia

**DOI:** 10.3390/ijerph18094803

**Published:** 2021-04-30

**Authors:** Nadeem Ali, Mohammad W. Kadi, Hussain Mohammed Salem Ali Albar, Muhammad Imtiaz Rashid, Sivaraman Chandrasekaran, Ahmed Saleh Summan, Cynthia A. de Wit, Govindan Malarvannan

**Affiliations:** 1Centre of Excellence in Environmental Studies, King Abdulaziz University, Jeddah 21589, Saudi Arabia; mimurad@kau.edu.sa (M.I.R.); scsekaran@kau.edu.sa (S.C.); asumman@kau.edu.sa (A.S.S.); 2Department of Chemistry, Faculty of Sciences, King Abdul Aziz University, Jeddah 21589, Saudi Arabia; mkadi@kau.edu.sa; 3Department of Community Health, Medical College, King Abdul Aziz University, Jeddah 21589, Saudi Arabia; hmalbar@kau.edu.sa; 4Department of Environmental Sciences, King Abdul Aziz University, Jeddah 21589, Saudi Arabia; 5Department of Environment Science, Stockholm University, Stockholm 11419, Sweden; Cynthia.deWit@aces.su.se; 6Toxicological Center, University of Antwerp, Antwerpen 2610, Belgium

**Keywords:** flame retardants, phthalates, PAHs, PCBs, car dust, human exposure

## Abstract

People may spend a significant amount of their daily time in cars and thus be exposed to chemicals present in car dust. Various chemicals are emitted from during car use, contaminating the car dust. In this study, we compiled published and unpublished data on the occurrence of phthalates, flame retardants (FRs), polycyclic aromatic hydrocarbons (PAHs), and polychlorinated biphenyls (PCBs) in Saudi car dust. Phthalates, a class of chemical commonly used as plasticizers in different car parts, were the major pollutants found in car dust, with a median value of ∑phthalates 1,279,000 ng/g. Among other chemicals, organophosphate flame retardants (OPFRs) were found to be between 1500–90,500 ng/g, which indicates their use as alternative FRs in the car industry. The daily exposure to Saudi drivers (regular and taxi drivers) was below the respective reference dose (RfD) values of the individual chemicals. However, the estimated incremental lifetime cancer risk (ILCR) values due to chronic exposure to these chemicals was >1 × 10^−5^ for taxi drivers for phthalates and PAHs, indicating that the long-term exposure to these chemicals is a cause of concern for drivers who spend considerable time in cars. The study has some limitations, due to the small number of samples, lack of updated RfD values, and missing cancer slope factors for many studied chemicals. Despite these limitations, this study indicates the possible range of exposure to drivers from chemicals in car dust and warrants further extensive studies to confirm these patterns.

## 1. Introduction

Several classes of chemicals, such as plasticizers and flame retardants (FRs), are used in the textiles, plastic, and rubber of vehicles to provide longevity, fulfil fire safety regulations, and prevent corrosion [1,2,3]. These chemicals are released from these products over time and distributed into the surroundings, such as the car interiors [1,2,3,4,5]. Regarding cars’ safety features, the consumer thinks about airbags, crumple zones, and seat belts [1]. However, most drivers and passengers are unaware of the hidden hazards that may pose a health risk unrelated to accidents [1,6]. Chemicals used in the floor coverings, electrical and electronic parts, seat cushions, and plastic parts to provide fire safety and longevity are released from these parts and accumulate in dust [1,4,5,7,8]. Drivers and passengers are exposed to this contaminated dust via inhalation, involuntary ingestion, and dermal contact [6].

According to the US Environmental Protection Agency (US EPA), indoor pollution is among the top five environmental risks to public health [9]. In Saudi Arabia, besides homes and offices, people spend a lot of time in cars for driving to work and leisure activities. There are domestic and commercial drivers, e.g., taxi drivers, who spend up to 10 h per day behind the wheel. During working hours, these drivers are exposed to contaminated dust, which affects their health because car dust contains many chemicals, e.g., FRs, phthalates, polycyclic aromatic hydrocarbons (PAHs), and PCBs [3,4,5,7,8,10,11]. In addition to being used as additives in polymers, some chemicals, e.g., PAHs, enter cars from the ambient environment during cross ventilation and due to dust tracked in from the outside on shoes [11]. A previous study reported cars as significant contributors to human indoor chemical exposure. Some chemicals were found to be five- to ten-fold higher in car interiors than in homes and offices [1]. The study reported that exposure to some chemicals during a 90 min drive was equivalent to exposure from eight hours of office work [1]. A high-level of exposure to these chemicals during driving presents health concerns. Many studies have shown that these chemicals are associated with health effects, such as endocrine disruption, reproductive health, and possibly carcinogenicity [12,13,14,15].

In this study, previously published data on phthalates, organophosphate flame retardants (OPFRs), polybrominated diphenyl ethers (PBDEs), new brominated FRs (BFRs), and PAHs were compiled [5,11,16], together with unpublished data on PCBs, are discussed in detail to study the levels and profiles of these chemicals in car dust in Jeddah, Saudi Arabia. In previous studies, the data on these chemicals were briefly mentioned but not discussed in detail. The compiled contaminant data were used to estimate the long-term exposure to drivers via car dust ingestion, dermal contact, and air inhalation and determine potential health risks through comparison with the estimated intake reference dose values.

## 2. Materials and Methods

Table 1 lists the chemicals and references for the publications included in the study. Data were compiled for phthalates, FRs, and PAHs from our earlier published studies [5,11,16], and new unpublished data on PCBs (CBs: 101, 118, 153, 138, 187, 180, 170) were also included.

### 2.1. Sampling and Instrumentation

Details about sampling and instrumentation are provided elsewhere [5,11,16]. In brief, dust samples were collected from the inside of cars (taxis and regular cars) (*n* = 15) from Jeddah, KSA. A vacuum cleaner was used to collect dust from vehicles. The car’s interior (dashboard, seats, and trunk) was vacuumed, excluding the floor, and before each sample, the vacuum cleaner was cleaned thoroughly to avoid any cross-contamination. To achieve homogenized samples, a 250 µm mesh was used to sieve all samples. The samples were extracted using solid-phase extraction, and gas chromatography-mass spectrometry (GCMS) was used for the quantitative analysis. Detailed instrumentation methods for BFRs, OPFRs, PAHs, phthalates, and PCBs are described elsewhere [5,11,16].

### 2.2. Health Risk Assessment Methodology

#### 2.2.1. Exposure Assessment via Dust Ingestion

The estimated daily intake (EDI) of contaminants from dust ingestion was estimated from Equation (1).
Estimated daily intake (ng/kg BW/day) = (C_n_ × I_R_/BW) × F_time_(1)
where C_n_ indicates the concentrations of chemicals in the dust (ng/g), I_R_ is the dust ingestion rate (200 mg/day for drivers), BW is body weight (70 kg body weight was considered for the adult drivers), and F_time_ is the fraction of time taxi drivers spend in the car (10 h, which is approximately 42% of one day). The outdoor environment of Saudi Arabia is dry, dusty, and hot most of the year. As a result, air conditioning (AC) is used inside cars throughout the year. AC use increases the airflow inside the vehicles and, consequently, increases the chance for drivers to be exposed to dust from inside the car. Therefore, we used a higher intake of dust for our calculations, i.e., 200 mg/day [11]. Most of the taxi services in Jeddah are run by foreign taxi drivers who live without their families and drive even longer shifts; therefore, we estimated a ten-hour working day. With the lack of knowledge on these chemicals’ bioaccessibility, we assumed 100% bioaccessibility for the EDI.

#### 2.2.2. Incremental Lifetime Cancer Risk (ILCR)

Long-term exposure to some of these chemicals is linked with carcinogenic risk. Therefore, in this study, incremental lifetime cancer risk (ILCR) was calculated using http://www.popstoolkit.com/tools/HHRA/Carcinogen.aspx (accessed on 10 March 2021) from Health Canada [17]. ILCR is one way of estimating the long-term exposure risk associated with exposure to chemicals. Equations (2)–(5) were used to calculate the ILCR.
(2)Inhalation dose =Cn × Pair × IRA × AFinh × Dhours × Ddays × Dweeks × DyearsBW × 365 × LE × 10^9
(3)$$\mathrm{Ingestion}\;\mathrm{dose}\;=\frac{C_{n}\;\times \;\mathrm{IRD}\;\times \;{\mathrm{AF}}_{\mathrm{GIT}}\;\times \;D_{\mathrm{hours}}\;\times \;D_{\mathrm{days}}\;\times \;D_{\mathrm{weeks}}\;\times \;D_{\mathrm{years}}}{\mathrm{BW}\;\times \;365\;\times \;\mathrm{LE}\;\times \;16}$$
(4)$$\mathrm{Dermal}\;\mathrm{dose}\;=\frac{C_{n}\;\times \;{\mathrm{SA}}_{H}\;\times \;{\mathrm{SL}}_{H}\;\times \;{\mathrm{AF}}_{\mathrm{skin}}\;\times \;\mathrm{EF}\;\times \;D_{\mathrm{hours}}\;\times \;D_{\mathrm{days}}\;\times \;D_{\mathrm{weeks}}\;\times \;D_{\mathrm{years}}}{\mathrm{BW}\;\times \;365\;\times \;\mathrm{LE}}$$
(5)ILCR = (Ingestion dose × SF Oral) + (Inhalation dose × SF Inhalation) + (Dermal dose × SF Dermal)

In the above equations, C_n_ represents the concentration (mg/kg) of the chemical in the dust. P_air_ is the concentration of particles in the air; 0.76 μg/m^3^ was used, which is recommended by US EPA [18] for typical conditions. IRA is the inhalation rate in m^3^/h, which was 0.658 for adults, and IRD is the ingestion rate of dust, which was considered 0.0002 kg/day for the drivers [17]. AF_GIT_, AF_Inh_, and AF_Skin_ are the absorption factors for the gastrointestinal tract, lungs, and skin, respectively. For this preliminary risk assessment [17], the value of 1 was used for AF_GIT_ and AF_Inh_, while for AF_Skin_, dermal absorption values from the literature were used, i.e., 0.02 for phthalates, 0.2 for PAHs, 0.001 for FRs, and 2 for PCBs [17,18,19,20,21]. SAH and SLH represent the exposed surface area (cm^2^) of the body and dust loading (kg/cm^2^/event) to the exposed area, respectively. We considered only the exposed hand area, i.e., 890 cm^2^, for taxi drivers, because this part of the body is most likely the most exposed to chemicals via contaminated dust [17,19]. According to Health Canada [17], hands have ten times greater loading of contaminated dust than other parts of the body; therefore, 0.0001 g/cm^2^/event was used for the calculations. This is because drivers touch surfaces of the cars with bare hands and then unknowingly touch their hands to their face; this hand-to-face behavior contributes greatly to the loading of contaminated dust to the body. EF represents the exposure frequency; we considered 1 h for these calculations. D_Hours_ is the hours of exposure per day, which was set at 10 h. D_Days_, D_Weeks_, and D_Years_ are the number of working days in a week (6 days), the number of working weeks in a year (48 weeks), and the number of years of exposure (25 years) for drivers, respectively. BW represents body weight, which was considered to be 70 kg. LE represents life expectancy, i.e., the number of years an average person is likely to live; we used 75 years in our study. SF represents the cancer slope factor for the chemicals, an upper bound on the increased cancer risk from a lifetime exposure to an agent; the SF is expressed as mg/kg/day. SF varies for each chemical and exposure route, e.g., inhalation, ingestion, and dermal contact. However, in the literature, SF values for many of these chemicals are not available for each exposure route. Therefore, when SF for separate exposure routes was not available, SF for ingestion was used for the ILCR.

## 3. Results and Discussion

### 3.1. The Presence of Different Chemical Groups in Car Dust

We compiled published and unpublished data on the occurrence of phthalates [5], polycyclic aromatic hydrocarbons [11], flame retardants [16], and PCBs in Saudi car dust. Phthalates were the major contaminants found in all car dust samples, followed by OPFRs, PAHs, BFRs, and PCBs (Table 2 and Figure 1). PCBs contributed the least to the chemical profile, which is why its contribution is not visible in Figure 1. The levels and profiles of each group of chemicals are discussed in detail below.

#### 3.1.1. Levels and Profiles of Phthalates and PCBs

Phthalates were the most significant class of chemicals found in the car dust samples (Table 2, Figure 1 and Figure 2). The levels of phthalates were multifold higher than other types of compounds and ranged between 85,600–3,330,000 ng/g of dust. The levels of phthalates found in these samples indicate high use of these chemicals as plasticizers in different car materials, e.g., large quantities of phthalates are added in compounded polyvinyl chloride (PVC) for molded use. The addition of phthalates in various parts of cars enhances the ability of automobile components to withstand high temperatures and makes them more resistant to degradation [1,2]. PVC coatings and components in vehicles also help prevent corrosion from water and weather elements. Flexible vinyl is also used in cars and trucks to make them lighter and more fuel-efficient.

DEHP is the preferred phthalate in a variety of PVC products, and it can be up to 30–45% by weight in most flexible PVC applications in cars [22]. DEHP was the primary phthalate found in all dust samples (Figure 2; Appendix A). DEHP contributed more than 80% of the total phthalate load in all but four dust samples, averaging an 83% contribution in the ∑phthalates profile (Appendix A). Phthalates, including DEHP, are additives in polymers like PVC, and with time, DEHP is released from the product due to volatilization. The volatilization process of phthalates from the treated product increases with high ambient temperature. DEHP is a medium category volatility plasticizer, and the volatilization increases 10,000-fold when temperature increases from 20 to 100 °C [1]. The outdoor temperature is generally high in Jeddah, KSA, reaching up to 45 °C in summer. In a parked car, the indoor temperature can reach 60 °C, which increases the volatilization of phthalates, including DEHP, from the treated materials [1]. Therefore, the high concentrations of DEHP in the studied car dust is not surprising. DIBP, DBP, and DOP were other important phthalates that contributed on average >3% to the total phthalate load.

PCB 118, 153, and 180 (Appendix A) were the major PCB congeners found in the car dust samples, though most of the detected PCBs were present in less than 50% of car dust samples. PCBs have been regulated since 1977 and banned globally due to the health effects associated with their exposure, but they are still present in the environment in declining levels. However, PCBs are persistent in the environment, and they bind with soil and sediments [8,10,23]. Therefore, they are still found in environmental samples at low concentrations, and studies have shown their levels are declining [8,10,23]. The low levels of PCBs in the present study indicate no recent use or exposure via recycling.

To the best of our knowledge, only two other studies have reported the presence of PCBs in car dust [8,10]. Concentrations of all PCB congeners in the present study were similar to those found in car dust samples from Kuwait and Pakistan, while they were much lower than those reported in Nigerian car dust [8,10].

#### 3.1.2. Levels and Profiles of FRs

The commercial production and use of Penta-BDE and Octa-BDE mixtures have been regulated since 2009 due to the widespread presence of these chemicals in different environmental media and their documented toxic properties [24,25,26]. Due to fire safety regulations, textiles, upholstery, plastic, and rubber materials are treated with FRs, and PBDEs were used widely in the past. FRs added to consumer products are released into the surrounding environments, as shown by their occurrence in car dust and other environmental compartments [4,10,16,24,25,26]. Due to the restriction and global banning of PBDEs, the use of emerging BFRs and OPFRs as alternatives has increased, e.g., DBDPE for Deca-BDE, BTBPE for Octa-BDE, and Firemaster 550 (a mixture of TBB, TBPH, and TPhP) for Penta-BDE [4,10,16,26]. Recent studies have shown the occurrence of these replacement FRs in the environment [25,26], including in the Arctic, as well as possible impacts on human health, creating a growing cause of concern [27,28,29,30,31].

The data on Saudi car dust shows that, except for BDE 209 and DBDPE, median levels were low for PBDEs and new BFRs, (Table 2, Figure 2 and Appendix A). The relatively high levels of BDE 209 and DBDPE compared with other BFRs are probably due to their higher production volumes and more extensive use than other BFRs in different car materials [4,16]. However, a few car dust samples showed high TBB and TBPH (Appendix A). In the same dust samples, TPhP levels were also increased, which indicates the use of Firemaster 550 as a possible source of these chemicals. Compared to BFRs, OPFRs in these car dust samples were more frequent and present at higher concentrations, especially chlorinated OPFRs (Table 2). TDCPP, TCPP, and TCEP were present in the car dust samples, with median levels of 2700, 1650, and 1200 ng/g, respectively (Table 2, Appendix A). Similar results have been reported in car dust samples from other countries [4]. Abdallah et al. [4] indicated that in some countries OPFRs are already preferred over other FRs for polypropylene and acrylonitrile butadiene styrene in instrument panels, textiles, polyurethane foam in the interior upholstery, and numerous electronics used inside of cars [4]. Studies have reported that chlorinated PFRs are more stable and resistant to biodegradation, which might also explain their higher occurrence [4,16,32].

#### 3.1.3. PAHs

The presence of PAHs, as found in Saudi car dust, was reported in other indoor microenvironments, namely household floor dust and AC filter dust, where a greater focus was given to other microenvironments [11]. BbF, Pyr, Phe, Flu, and Fln were the main PAHs detected in car dust, with median levels >200 ng/g of dust (Table 2, Figure 2and Appendix A). BaP, the PAH with the highest carcinogenic potency, was present at a median concentration of 145 ng/g of dust. Studies have linked high molecular weight PAHs with an increased risk of carcinogenicity [33]. In the literature, the BaP equivalent (BaPE) is reported as an index to evaluate the toxicity of PAHs in each environment using Equation (6) [11]:BaPE = BaA × 0.06 + (Bbf + BkF) × 0.07 + BaP + DahA × 0.6 + IcdP × 0.08(6)

The BaPE for the Saudi car dust samples ranged between 5 and 630 ng/g, with a median value of 180 ng/g—double the BaPE found in Kuwaiti car dust [11]. This higher BaPE value indicated a higher carcinogenic risk from PAHs to Saudi drivers than Kuwaiti drivers. PAHs are categorized according to their aromatic rings, i.e., two to six aromatic rings. The considerable contribution of PAHs in Saudi car dust came from those with three (36%), four (30%), and five (25%) rings, while PAHs with six rings contributed much less (4%) (Figure 3). The low molecular weight PAHs are more volatile, which explains their relatively lower contribution in the PAH profile (2–3 aromatic rings) (~40%) than high molecular weight PAHs (4–6 aromatic rings) (~60%). The compositional pattern indicated that car dust is an important source of human exposure to low and high molecular weight PAHs (Figure 3). Therefore, settled dust is an important exposure source of toxic PAHs via involuntary dust ingestion and dermal contact. The source apportionment diagnostic ratios BghiP/BaP (1.18), BFs/BghiP (1.82), IcdP/(IcdP + BghiP) (0.31), BaA/(BaA + Chr) (0.51), Flu/(Flu + Pyr) (0.49), and Phe/Ant (4.45) indicated both petrogenic and pyrogenic pollution were the main primary sources for the occurrence of these PAHs in car dust samples [11]. The city of Jeddah is densely populated with high per capita petroleum-based energy consumption [11]. Therefore, the production and combustion of many petroleum products might significantly contribute to the release of PAHs into the ambient environment. Thus, the PAH levels in car dust are probably due to contaminated outdoor air during cross ventilation and dust tracked in with shoes.

### 3.2. Human Risk Assessment

As stated above, several studies have reported that exposure to many of the analyzed chemicals in car dust are linked to various health effects. The most prominent compound in the study—DEHP—is an endocrine disruptor and has reported carcinogenic properties [34]. Rowdhwal and Chen [35] reviewed the toxicity of DEHP and discussed the health implications linked to its exposure, e.g., testicular, ovarian, endometrial, neuro-, cardio-, and hepatotoxicity. Exposure to phthalates and their metabolites has been shown to cause potential health implications for the liver and kidneys, which lead to low BMI, increased organ weight, and endocrine disruption as well as teratogenic, developmental, and reproductive outcomes [36]. Like phthalates, exposure to OPFRs has also been linked to several health effects. Several studies are available in the literature that show that OPFRs are endocrine disruptive and exhibit carcinogenic properties [14,37,38]. PBDEs are found in human samples (serum, milk, fat, hair, etc.), and various health problems are associated with exposure to these chemicals and their metabolites, such as neurobehavioral and reproductive disorder, thyroid hormone disruption, etc. [39,40,41,42]. Based on the limited evidence of carcinogenicity in humans and animals, the International Agency for Research on Cancer (IARC) and the US EPA have classified PBDEs as a Group 3 and Group D carcinogen (not classifiable as to its carcinogenicity to humans), respectively [40]. Data on the health implication of new BFRs is lacking in the literature, and only a few studies are available that are focused on the toxicity of these chemicals [26,43]. A range of various health effects is associated with exposure to PCBs, especially in occupational settings. Chronic exposure to PCBs has been linked to cardiovascular, gastrointestinal, neurological, musculoskeletal, carcinogenic, reproductive, and hepatic effects as well as endocrine disruption [44]. PAHs are classified as probable carcinogens (Group 1, 2A, and 2B) to humans by the IARC [45], and some PAHs are also mutagens and teratogens and therefore pose a serious threat to the health and the well-being of humans [15]. Exposure to a mixture of PAHs is associated with a series of health issues, e.g., increased risk of lung, skin, bladder, and gastrointestinal cancers. Exposure to individual PAHs is not well studied in humans, with most of the studies focused on the exposure to mixtures of PAHs in humans [15].

Various exposure scenarios using the fifth percentile (low-end exposure), mean, median, and 95th percentile (high-end exposure) concentrations were calculated for daily exposure via dust ingestion for taxi drivers and regular drivers (Appendix A). Low-end exposure using fifth percentile was used for those drivers exposed to fewer chemicals from their cars, as they spend less time on car seats and also keep their vehicles clean with regular car service. While, high-end exposure using 95th percentile was more relevant to those drivers who spend a lot of time driving and also do not regularly clean their cars. For both taxi and regular drivers, the estimated exposure levels (Appendix A) for most of the chemicals were multifold lower than their reference dose (RfD). However, calculated exposure to BaP for taxi drivers was more than the virtually safe amount (Appendix A). Many of these RfD values are based on old toxicological studies, which need to be updated to understand the current exposure threat from these chemicals. The estimated exposure dose is less than the corresponding RfD for daily exposure from cars. However, if exposure pathways, i.e., indoor household dust, indoor and outdoor air, and food intake, are included, total exposure might be a cause of concern. Phthalates contributed most to the daily exposure (~99%), primarily via dust ingestion; among phthalates, DEHP was the paramount contributor (Figure 4, Appendix A). All other chemical groups contributed only 1% of the exposure load (Figure 4). Although many of the individual chemicals were below RfD values, they might have similar health effects. The synergetic impact on overall health from such complex mixtures is not well studied. Therefore, it is crucial to monitor chemicals in indoor environments, particularly phthalates, which are reported at high levels in different environmental media.

To study the potential long-term cancer risk of exposure of Saudi taxi and regular drivers to car dust, the ILCR was calculated for the different analytes. The probabilistic chronic exposure assessment was highest via dust ingestion, followed by dermal contact, and inhalation (Table 3). The US EPA recommended safe limit for long-term cancer risk is below 1 × 10^−5^ [46]. For most of the studied chemicals, the estimated ILCR was below 1 × 10^−5^, which indicates a limited carcinogenic risk to Saudi drivers from those chemicals. However, for some substances, namely ƩPAHs and Ʃphthalates, the estimated ILCR was >1 × 10^−5^ for Saudi taxi drivers. This indicates that taxi drivers, which spend a substantial amount of time in car seats, have a potential cancer risk from long-term exposure to these chemicals from cars. As shown by Figure 5, phthalates and PAHs are the two major groups of chemicals that could pose long-term health risks due to carcinogenicity to Saudi drivers from car dust. Figure 5 shows that ∑phthalates and ∑PAHs contribute 55% and 44%, respectively, to the ILCR profile compared to just 1% from the other three classes of chemicals. Although OPFR levels are much higher in analyzed car dust than those of the PAHs, chronic exposure to PAHs via car dust exposure is of greater concern due to the higher carcinogenicity associated with them. One of the noteworthy points about ILCR calculations is the choice of used cancer slope factor; it can significantly affect the estimated risk assessments and, as a result, risk management decisions [47]. Cancer slope factors vary significantly in different jurisdictions, e.g., the cancer slope factor (per mg/kg BW/day) used for BaP is 7.3 (USA, New Zealand), 2.3 (Canada), 0.46 (WHO Drinking-water Quality), and 9.03 (California) [47]. This study employed the oral cancer slope factor for BaP (ƩPAHs) recommended by Health Canada (i.e., 2.3 mg PAH/kg BW/day). If the cancer slope factor of California (9.03 mg kg BW/day) were used, the risk estimation would have increased significantly.

The ILCR calculated for ∑phthalates (using both median and maximum concentrations), ∑PAHs (using maximum levels), and ∑all chemicals via car indoor dust exposure for taxi drivers were >1 × 10^−5^, signifying that the carcinogenic risk from long-term exposure is a cause of concern for their health [46]. Lee et al. [48] reported a positive correlation between oxidative stress and high levels of DEHP metabolites in urine samples from Riyadh, Saudi Arabia. They suggested that indoor dust was one of the main exposure pathways. Limited to no information is available on the production and use of DEHP in Saudi Arabia, which is a cause of concern. According to EU REACH legislation, DEHP is a Category 1B reprotoxic substance (The European Chemicals Agency), and its use is highly regulated. The calculated ILCR assessment in this study showed that taxi drivers in the studied area get exposed to chemicals via car dust. This exposure is significant to the extent that it might impact their health.

## 4. Conclusions

The high concentrations of DEHP in car dust are a cause of concern for drivers, especially those who spend long hours behind the wheel. The levels of OPFRs and new BFRs were higher than the regulated and banned PBDEs. PCBs were detected in low concentrations in some dust samples. The daily exposure to all individual chemicals was below RfD values, indicating no risk of health effects from acute exposure for the individual chemicals. However, chronic exposure to phthalates and PAHs was of great concern due to their carcinogenicity. The estimated ILCRs for both ∑phthalates and ∑PAHs were above 1 × 10^−5^ for Saudi taxi drivers. This signifies that long-term exposure to these chemicals can negatively impact drivers’ health. Large-scale studies based on dust samples from more cars are needed to understand the clear picture of these chemicals in car dust. Another significant problem is the lack of updated toxicological studies on the studied chemicals. Thus, no RfD or cancer slope factors are available for many of the studied chemicals; this makes it challenging to estimate the risk accurately. Updated RfDs, cancer slope factors, and better data on the bioavailability of chemicals are required for improving risk assessments. However, even with these limitations, this study indicates that drivers are exposed to various chemicals from car dust and warrants further extensive studies of different types of cars and other vehicles such as trucks and busses.

## Figures and Tables

**Figure 1 ijerph-18-04803-f001:**
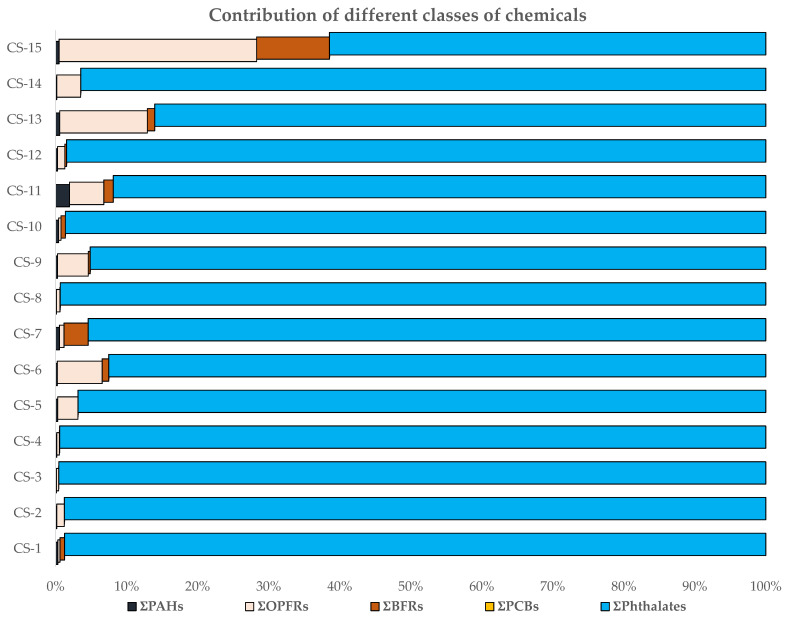
Contribution of each group of chemicals in the dust chemical profile of the 15 individual car dust samples. Values on the *X*-axis are given in ng/g of dust, while the *Y*-axis represents each car dust sample’s ID, CS represent “car sample”.

**Figure 2 ijerph-18-04803-f002:**
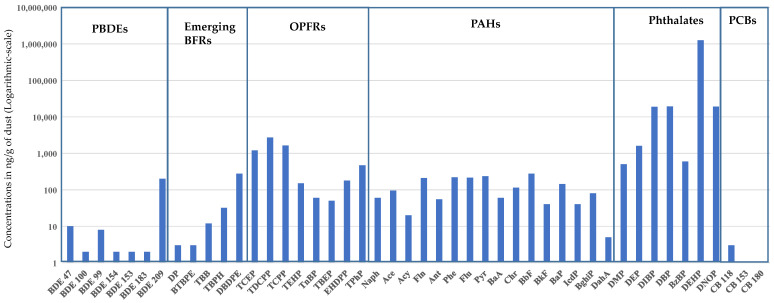
Profile of individual analyzed chemicals in Saudi car dust. The concentration (ng/g) on *y*-axis is in a logarithmic scale.

**Figure 3 ijerph-18-04803-f003:**
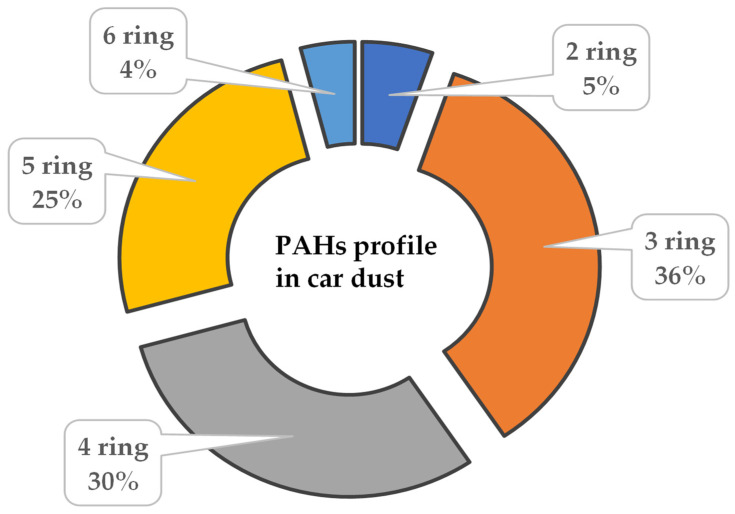
Contribution of ring-based PAHs in car dust.

**Figure 4 ijerph-18-04803-f004:**
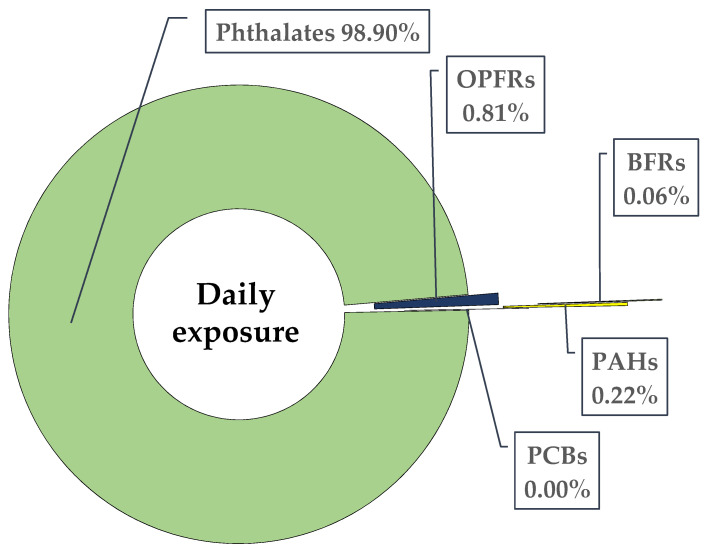
Chemical profile (%) of daily exposure from car dust based on median values (ng/kg BW/day).

**Figure 5 ijerph-18-04803-f005:**
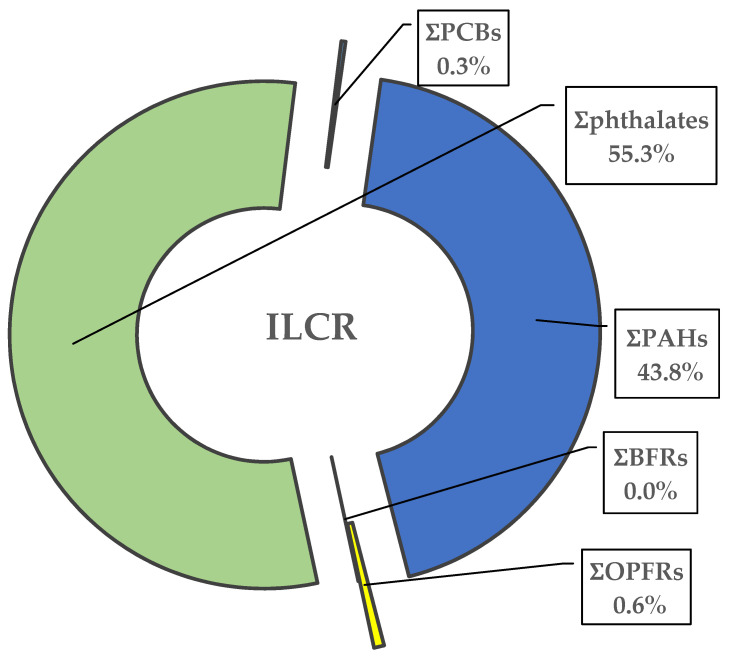
Chemical profile (%) of the estimated ILRC from car dust.

**Table 1 ijerph-18-04803-t001:** Names and abbreviations of chemicals included and references to the published data compiled for this study.

Chemicals	Abbreviations
Brominated flame retardants (BFRs) [16]	
Polybrominated diphenyl ethers (PBDEs)Congeners (28, 47, 99, 100, 153, 154, 183, and 209)	BDE 28, BDE 47, BDE 99, BDE 100, BDE 153, BDE 154, BDE 183, BDE 209
Decabromodiphenylethane	DBDPE
1,2-bis(2,4,6-tribromophenoxy) ethane	BTBPE
Bis(2-ethylhexyl)-3,4,5,6-tetrabromophthalate	TBPH
2-ethylhexyl-2,3,4,5-tetrabromobenzoate	TBB
Organophosphate flame retardants (OPFRs) [16]	
Tris-(2-chloroethyl)-phosphate	TCEP
Tris-(1,3-dichloro-isopropyl)-phosphate	TDCPP
Tris-(1-chloro-2-propyl)-phosphate	TCPP
Triphenyl phosphate	TPhP
Tris(2-ethylhexyl) phosphate	TEHP
Tri-n-butyl phosphate	TnBP
Tris(2-butoxyethyl) phosphate	TBEP
2-ethylhexyl-diphenyl phosphate	EHDPP
Phthalates [5]	
Dimethyl phthalate	DMP
Diethyl phthalate	DEP
Benzyl butyl phthalate	BzBP
Di-n-butyl phthalate	DBP
Di-isobutyl phthalate	DIBP
Bis(2-ethylhexyl) phthalate	DEHP
Di-n-hexyl phthalate	DNHP
Dicyclohexyl phthalate	DCHP
Di-n-octyl phthalate	DNOP
Polycyclic aromatic hydrocarbons (PAHs) [11]	
Acenaphthene	Ace
Acenaphthylene	Acy
Anthracene	Ant
Benz(a)anthracene	BaA
Benzo(a)pyrene	BaP
Benzo(b)fluoranthene	BbF
Benzo(g,h,i)perylene	BghiP
Benzo(k)fluoranthene	BkF
Chrysene	Chr
Dibenz (a, h) anthracene	DahA
Fluoranthene	Flu
Indeno(1,2,3-cd) pyrene	IcdP
Naphthalene	Naph
Phenanthrene	Phe
Pyrene	Pyr
Polychlorinated biphenyls (PCBs)	
PCBs (101, 118, 153, 138, 187, 180, 170)	CB101, CB118, CB153, CB153, CB138, CB187, CB180, CB170

**Table 2 ijerph-18-04803-t002:** Descriptive statistics of analyzed chemicals in Saudi car dust (*n* = 15). All values are given in ng/g of dust.

Chemical Group	Analytes	Detection Frequency (%)	Mean ± STD	Median (Mini-Max)
**PBDEs**	BDE 47	87	21 ± 50	10 (LOQ-200)
	BDE 100	53	7 ± 17	2 (LOQ-700
	BDE 99	93	40 ± 100	8 (LOQ-400)
	BDE 154	20	3 ± 6	2 (LOQ-25)
	BDE 153	33	6 ± 9	2 (LOQ-30)
	BDE 183	7	3 ± 5	2 (LOQ-20)
	BDE 209	100	5900 ± 11,650	200 (15–35,500)
**Emerging BFRs**	BTBPE	73	10 ± 20	3 (LOQ-70)
	TBB	100	760 ± 2300	12 (3–8700)
	TBPH	93	200 ± 550	32 (LOQ-2150)
	DBDPE	100	850 ± 1500	275 (45–6020)
**OPFRs**	TCEP	100	7,020 ± 13,850	1200 (30–52,300)
	TDCPP	100	8,850 ± 13,500	2700 (100–45,600)
	TCPP	100	16,250 ± 28,700	1650 (100–92,000)
	TEHP	80	195 ± 205	150 (<LOQ-850)
	TnBP	100	540 ± 1670	60 (20–6550)
	TBEP	47	1650 ± 4000	50 (<LOQ-12,500)
	EHDPP	100	1050 ± 2300	180 (60–9000)
	TPhP	100	786 ± 1100	470 (40–4150)
**PAHs**	Naph	80	60 ± 50	60 (<LOQ-135)
	Ace	100	110 ± 45	95 (65–220)
	Acy	67	25 ± 30	20 (<LOQ-120)
	Fln	93	495 ± 520	210 (<LOQ-1520)
	Ant	73	65 ± 45	55 (15–175)
	Phe	100	280 ± 200	220 (35–580)
	Flu	100	395 ± 380	215 (65–1480)
	Pyr	100	410 ± 315	235 (85–520)
	BaA	100	75 ± 60	60 (25–280)
	Chr	100	125 ± 75	115 (25–250)
	BbF	80	340 ± 385	275 (<LOQ-1570)
	BkF	67	80 ± 100	40 (<LOQ-370)
	BaP	87	145 ± 115	145 (<LOQ-460)
	IcdP	73	65 ± 60	40 (<LOQ-175)
	BghiP	53	85 ± 90	80 (<LOQ-250)
	DahA	20	7 ± 15	5 (<LOQ-45)
**Phthalates**	DMP	40	1700	500 (110–10,500)
	DEP	100	2600	1600 (690–8700)
	DIBP	100	119,000	18,900 (4400–831,000)
	DBP	100	46,000	19,400 (4290–356,000)
	BzBP	66	1700	600 (260–12,600)
	DEHP	100	1,170,000	1,250,000 (62,600–2,446,000)
	DNOP	100	47,800	19,000 (3570–319,000)
**PCBs**	CB 118	53	2 ± 2	2 (<LOQ-6)
	CB 153	40	1 ± 2	<0.2 (<LOQ-8)
	CB 180	33	1 ± 1	<0.2 (<LOQ-3)

**Table 3 ijerph-18-04803-t003:** The potential cancer risk assessment for taxi and regular drivers via dust exposure. **Bold** values are considered as cause of concern.

Chemical Class	Exposure Group	Stat	Ingestion Dose	Inhalation Dose	Dermal Dose	ILCR
ƩPAHs	Taxi driver	Median	1.5 × 10^−06^	5.9 × 10^−11^	2.1 × 10^−06^	8.2 × 10^−06^
		Max	3.9 × 10^−06^	1.6 × 10^−10^	5.5 × 10^−06^	**2.2 × 10^−05^**
	Regular driver	Median	2.9 × 10^−07^	1.2 × 10^−11^	4.2 × 10^−07^	1.7 × 10^−06^
		Max	7.7 × 10^−07^	3.1 × 10^−11^	1.1 × 10^−06^	4.3 × 10^−06^
ƩPhthalates	Taxi driver	Median	6.5 × 10^−04^	2.6 × 10^−08^	9.3 × 10^−05^	**1.0 × 10^−05^**
		Max	1.6 × 10^−03^	6.5 × 10^−08^	2.3 × 10^−04^	**2.6 × 10^−05^**
	Regular driver	Median	1.3 × 10^−04^	5.2 × 10^−09^	1.9 × 10^−05^	2.1 × 10^−06^
		Max	3.4 × 10^−04^	1.3 × 10^−08^	4.6 × 10^−05^	5.2 × 10^−06^
ƩOPFRs	Taxi driver	Median	5.3 × 10^−06^	2.1 × 10^−10^	7.6 × 10^−07^	1.2 × 10^−07^
		Max	4.6 × 10^−05^	1.8 × 10^−09^	6.6 × 10^−06^	1.1 × 10^−06^
	Regular driver	Median	1.1 × 10^−06^	4.3 × 10^−11^	1.5 × 10^−07^	2.4 × 10^−08^
		Max	9.2 × 10^−06^	3.7 × 10^−10^	1.3 × 10^−06^	2.1 × 10^−07^
ƩBFRs	Taxi driver	Median	4.1 × 10^−07^	1.6 × 10^−11^	2.9 × 10^−09^	2.9 × 10^−10^
		Max	1.6 × 10^−05^	6.4 × 10^−10^	1.1 × 10^−07^	1.1 × 10^−08^
	Regular driver	Median	8.1 × 10^−08^	3.3 × 10^−12^	5.8 × 10^−10^	5.7 × 10^−11^
		Max	3.2 × 10^−06^	1.3 × 10^−10^	2.3 × 10^−08^	2.2 × 10^−09^
ƩPCBs	Taxi driver	Median	1.5 × 10^−09^	6.1 × 10^−14^	2.2 × 10^−08^	5.1 × 10^−08^
		Max	4.1 × 10^−09^	1.6 × 10^−13^	5.8 × 10^−08^	1.4 × 10^−07^
	Regular driver	Median	3.1 × 10^−10^	1.2 × 10^−14^	4.4 × 10^−09^	1.0 × 10^−08^
		Max	8.1 × 10^−10^	3.3 × 10^−14^	1.2 × 10^−08^	2.7 × 10^−08^
ƩChemicals	Taxi driver	Median	-	-	-	**2.3 × 10^−05^**
		Max	-	-	-	**4.9 × 10^−05^**
	Regular driver	Median	-	-	-	3.8 × 10^−06^
		Max	-	-	-	9.8 × 10^−06^

## Data Availability

All of the data are presented in the article and associated Appendix A.

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
