# Peer review of "Semi-Volatile Organic Compounds in Car Dust: A Pilot Study in Jeddah, Saudi Arabia"

_ijerph, 2021, doi:10.3390/ijerph18094803_

Round 1

Reviewer 1 Report

Comments to ijerph-1188448:

The study compiled some published and unpublished data of various air pollutants of car dust in Saudi Arabia. It is very interesting and important to evaluate the health risks for the residents in Saudi Arabia. The paper can be published after the major revision. See the specific comments below:

The authors claimed that this is the pilot study in Saudi Arabia. They should have compared the results with some other regions around the world and specified the difference of the car dust pollution and health risks for the local residents.

Materials and methods: The authors should detailed describe the data source. How many samples? from which published or unpublished source? Too little information can be found in the Table 1 and suggest to providing more database information.

Line 77: Number of samples collected (n = 15), including taxis and ordinary cars?

Line 85: and Line 242: Please check if there is a problem with the title format?

Line 90: The reference 11 was based on Polycyclic aromatic hydrocarbons (PAHs) in indoor dust samples from Cities the indoor dust samples, could you choose one about the car dust data reference?

Line 116: Hands are the most exposed part of chemicals via contaminated dust, could you add more references?

Line 147: Compared with other studies to confirm the reliability of your data.

Line 173: You can add some references to support your point.

Line 221: The reference 11 was based on the indoor environment, could you choose other data about outdoor?

Line 239: Could you add several studies to verify the PAH levels are related to dust tracked in on shoes?

Line 324: What is the sample size of regular drivers and taxi drivers?

Author Response

The study compiled some published and unpublished data of various air pollutants of car dust in Saudi Arabia. It is very interesting and important to evaluate the health risks for the residents in Saudi Arabia. The paper can be published after the major revision. See the specific comments below:

The authors claimed that this is the pilot study in Saudi Arabia. They should have compared the results with some other regions around the world and specified the difference of the car dust pollution and health risks for the local residents.

Thanks for taking out time to review the manuscript and for your remarks. Necessary changes have been made in the text as per the reviewer comments. There are few studies in literature that have focused on these chemicals in car dust and we have mentioned those studies in the text with references.

Materials and methods: The authors should detailed describe the data source. How many samples? From which published or unpublished source? Too little information can be found in the Table 1 and suggest to providing more database information.

Agreed and we have provided these information in “Sampling and Instrumentation” subchapter. We have mentioned the number of samples and references for the studies.

Line 77: Number of samples collected (n = 15), including taxis and ordinary cars?

Yes from cars which included some taxis and ordinary cars, we have mentioned this in the revised manuscript.

Line 85: and Line 242: Please check if there is a problem with the title format?

We changed line 85 to “Health Risk Assessment methodology”.

Line 90: The reference 11 was based on polycyclic aromatic hydrocarbons (PAHs) in indoor dust samples from Cities the indoor dust samples, could you choose one about the car dust data reference?

There is a large variation in the literature regarding involuntary intake of dust for adults. In our previous studies we used high intake of dust for adults due to dry and dusty nature of ambient environment of KSA. In the provide reference PAHs are reported in three microenvironments (home, AC filter, and cars dust).

Line 116: Hands are the most exposed part of chemicals via contaminated dust, could you add more references?

Provided 2 references and also explained further with “This is because drivers touch surfaces of the cars with bare hand and then unknowingly touches those hands to the face, this way hand-to-face behavior contribute greatly in the loading of contaminated dust to the body.”

Line 147: Compared with other studies to confirm the reliability of your data.

We have provided references in following sentences.   

Line 173: You can add some references to support your point.

We added another couple of references for this sentence. 

Line 221: The reference 11 was based on the indoor environment, could you choose other data about outdoor?

In the provide reference [11] PAHs are reported in three microenvironments (home, AC filter, and cars dust).

Line 239: Could you add several studies to verify the PAH levels are related to dust tracked in on shoes?

This phenomenon is similar to house dust, where tracked in dust with shoes is considered as one of the major sources of chemicals in indoors. There are not many studies available on PAHs in car dust in literature, this also make this study important.

Line 324: What is the sample size of regular drivers and taxi drivers?

We did not collect exact number of taxis in this study but taxis in the city are similar to the most common cars on the road even used by the regular drivers, so we combine both types of cars for calculations since samples size is already small. This study is important as it is showing many chemicals which are not available in literature e.g., PAHs. This study also provides us baseline data for future long scale study which we are already planning based on these results.   

Reviewer 2 Report

In the present work the health effects caused by exposure to different types of organic chemical compounds present in car interior dust were investigated. The work involved automobiles in Arabia where the particular environmental conditions favor this problem.

The topic is of great scientific importance also because the problem of indoor pollution is very relevant and not yet well known.. The work  is certainly very interesting, well structured and logically organized. It can be accepted with some minor revisions.

Fig. 1 the blue  colors used to indicate the different contributions of PAHs and Phthalates are almost the same and I cannot discriminate the two contributions. Also, in the figure1 I don't see the yellow color that indicates the Ć©PCBs; I assume it's because it's content is very low, as also reported in the text. This point should be explained in the caption of the figure.

Line 112: can you explain what the other indoor microenvironments are?

Lines 117-118. could you explain this point better? For example, why are the hands more exposed to dust than the face?

Line 156: authors write that "DEHP was the primary 1 phthalate found in all dust samples (Figure S1)."  Does Figure S1  correspond to CS1 or to Fig. 1?.However from figure 1 I cannot understand that DEHP is the most important phthalate. It is not shown in the caption. I see it in Fig.2.

Line 188: “These chemicals are released..” Are these chemicals the BDE?? Not clear to me.

Line 208: “in a confined area of cars”. This last sentence is not very clear.

Lines 272-274: Please can you  explain more  the application of the  percentile with the different levels of exposure?

Lastly, I believe that the English form needs to be improved as it is not adequate to the importance of the work and needs a thorough review. There are several repetitions and the sentences are often too long. I am not a native speaker but I have tried to improve the English form in various parts of the work.I  invite the authors to consider my corrections that I report in the PDF file.

Author Response

In the present work the health effects caused by exposure to different types of organic chemical compounds present in car interior dust were investigated. The work involved automobiles in Arabia where the particular environmental conditions favor this problem.

The topic is of great scientific importance also because the problem of indoor pollution is very relevant and not yet well known. The work is certainly very interesting, well-structured and logically organized. It can be accepted with some minor revisions.

Fig. 1 the blue colors used to indicate the different contributions of PAHs and Phthalates are almost the same and I cannot discriminate the two contributions. Also, in the figure1 I don't see the yellow color that indicates the Ć©PCBs; I assume it's because its content is very low, as also reported in the text. This point should be explained in the caption of the figure.

Thanks for taking out time to review the manuscript and for your remarks. Necessary changes have been made in the text as per the reviewer comments. Figure 1 is revised with different colour scheme. Yes you are right that contribution of PCBs is not visible due to low concentrations of analysed PCBs and we have mentioned his is the text in revised manuscript. 

Line 112: can you explain what the other indoor microenvironments are?

We restructured the sentence to name those indoor microenvironments.

Lines 117-118. Could you explain this point better? For example, why are the hands more exposed to dust than the face?

Agreed and we explained it further with “This is because drivers touch surfaces of the cars with bare hand and then unknowingly touches those hands to the face, this way hand-to-face behavior contribute greatly in the loading of contaminated dust to the body.”

Line 156: authors write that "DEHP was the primary 1 phthalate found in all dust samples (Figure S1)."  Does Figure S1 correspond to CS1 or to Fig. 1? However from figure 1 I cannot understand that DEHP is the most important phthalate. It is not shown in the caption. I see it in Fig.2.

Figure S1 is provided in supplementary information. In revised manuscript we mentioned Figure 2 and Figure S2 to support the sentence. 

Line 188: “These chemicals are released...” Are these chemicals the BDE?? Not clear to me.

Changed to “Added FRs in the consumer products”.

Line 208: “in a confined area of cars”. This last sentence is not very clear.

Agreed and sentence is restructured in revised manuscript. 

Lines 272-274: Please can you explain more the application of the percentile with the different levels of exposure?

Agreed and we have explained it further about why we used 5th and 95th percentiles for exposure calculations.

Lastly, I believe that the English form needs to be improved as it is not adequate to the importance of the work and needs a thorough review. There are several repetitions and the sentences are often too long. I am not a native speaker but I have tried to improve the English form in various parts of the work. I invite the authors to consider my corrections that I report in the PDF file.

We improved the language according to your suggestions and believe that on revised manuscript language is markedly improved.   

Round 2

Reviewer 1 Report

The paper has been well revised. I think it can be accepted for publication now.